# High-Fidelity Image Synthesis from Pulmonary Nodule Lesion Maps using Semantic Diffusion Model

**Xuan Zhao**                                                             xz1919@imperial.ac.uk
**Benjamin Hou**                                                         bh1511@imperial.ac.uk
*Department of Computing, Imperial College London, London, UK*

## Abstract

Lung cancer has been one of the leading causes of cancer-related deaths worldwide for years. With the emergence of deep learning, computer-assisted diagnosis (CAD) models based on learning algorithms can accelerate the nodule screening process, providing valuable assistance to radiologists in their daily clinical workflows. However, developing such robust and accurate models often requires large-scale and diverse medical datasets with high-quality annotations. Generating synthetic data provides a pathway for augmenting datasets at a larger scale. Therefore, in this paper, we explore the use of Semantic Diffusion Models (SDM) to generate high-fidelity pulmonary CT images from segmentation maps. We utilize annotation information from the LUNA16 dataset to create paired CT images and masks, and assess the quality of the generated images using the Fréchet Inception Distance (FID), as well as on two common clinical downstream tasks: nodule detection and nodule localization. Achieving improvements of 3.96% for detection accuracy and 8.50% for $AP_{50}$ in nodule localization task, respectively, demonstrates the feasibility of the approach.

**Keywords:** controlled image synthesis, lung nodules, semantic diffusion model

## 1. Introduction

Accurate detection and localization of pulmonary nodules using Computed Tomography (CT) is one of the main ways to perform early diagnosis of lung cancer. Deep learning has aided the diagnosis of lung cancer since its emergence. However, current methods for detecting lung nodules typically only predict their centers, while the size of the nodules, a critical diagnostic criterion, is often overlooked. The volume of a nodule can be used to differentiate between benign and malignant nodules, with larger volumes often indicating malignancy. Additionally, changes in nodule volume over time can be used to assess treatment response or disease progression (Gavrielides et al., 2009). Lung nodules are often quite small, they can exhibit a wide range of shapes that vary drastically with different semantic features. Gradient-based inpainting methods, such as Poisson blending, are unreliable when the nodule volume is too small. On the other hand, simple cut-and-paste techniques can introduce spatial discontinuity. In recent literature, diffusion models have been proven to be capable of generating very realistic images (Yang et al., 2022), which is particularly useful as a data augmentation method for medical imaging tasks (Chen et al., 2022). In this paper, we leverage Semantic Diffusion Model (SDM) (Wang et al., 2022) to synthesize high-fidelity pulmonary CT images from segmentation masks containing lung nodules. Our proposed method permits controlled synthesis of nodule shape and size whilst maintaining image quality and diversity. The results obtained demonstrate the beneficial performance of our pipeline in subsequent downstream tasks.

## 2. Data and Method

Our method utilizes the LUNA16 dataset (Setio et al., 2017), a subset of the publicly available LIDC-IDRI dataset, which contains 888 chest CT scans and 1186 marked nodules. For all experiments, the CT window is set between [-1000,400], and operations are performed in 2D. Slices are considered only if they contain lung structure, as nodules do not exist outside these regions. To create nodule masks, the nodules are first cropped spherically based on the centroid and diameter information from the provided annotations. A manual OTSU threshold is then applied to each cropped region-of-interest to get the final masks. The intensities of the cropped pixels are clustered using the K-Means algorithm with two centers, and the threshold is selected as the average of these centers. Additionally, a 'body mask' is also generated, which comprises the entire patient's body. Each slice is intensity thresholded at 127, followed by a morphological hole fill process. The largest connected region is then selected. The final mask is then composed of the structures in this order; background, left lung, right lung, trachea, body mask, and nodule if one is present.

Our data preprocessing method above results in 1139 slices with nodules and 128059 slices without nodules. For all experiments, the training and testing sets are divided by patient ID to ensure no data cross-contamination. Specifically, 744 patients were used for training, while 144 patients were used for testing. As nodule-free slices would greatly outnumber slices with nodules (almost 1 in 100), only 1 in 4 (empirically selected) nodule-free slices are selected to train the generative models, as well as subsequent downstream tasks. SDM and SPADE (Park et al., 2019), a previous state-of-the-art method, are trained and used to generate synthetic 2D pulmonary CT slices; 1000 slices with nodules, and another 1000 that are nodule-free. Two downstream tasks, namely nodule detection and localization, are then trained with a mixture of synthetic and real samples. SDM was trained with an image size of 256x256 (reduced due to resource availability) and a batch size of 2. Training took approximately 2 days for 100,000 steps, using the AdamW optimizer with an initial learning rate of 1e-4. SPADE was trained with an image resolution of 512x512 and a batch size of 16. Training took approximately 7 hours for 20 epochs, using the Adam optimizer with an initial learning rate of 1e-4. All experiments were conducted on a machine with an NVIDIA A6000 GPU.

## 3. Experiments and Result

All experiments are run for 10-folds with the synthetic images in the test fold being excluded if there are any, and significance of accuracy/$AP_{50}$ is confirmed by Wilcoxon rank-sum test as shown in the p-value column. Table 1 shows the relevant metrics of two models before and after adding diffusion-generated images in the training set. For nodule detection task (determining whether a cropped $32 \times 32$ patch is nodule or non-nodule), a SE-ResNet (Hu et al., 2019) was trained. Table 1 shows that adding SDM-generated images increases both the accuracy and F1 score, as well as lower the standard deviation, when compared to the baseline and SPADE. For nodule localization task, a Faster R-CNN model (Ren et al., 2016) was trained for detecting the location/bounding boxes of nodules on a 2D slice. The model trained with the additional SDM-generated images outperformed both the baseline and the SPADE-image-trained model in AP and AR in all selected IoU test points. Figure 1A shows samples generated by SDM and SPADE, and Figure 1B shows

example downstream nodule localizations. Finally, the FID scores for SDM among nodule and non-nodule diffusion-generated images are 80.820 and 84.494, respectively, and the FID scores of those for SPADE-generated images are 186.609 and 147.451.

|  | Accuracy (%) | Precision (%) | Rec./Sen. (%) | Specificity (%) | F1 | p-value |
|---|---|---|---|---|---|---|
| I,A | $85.76 \pm 1.69$ | $\mathbf{85.63 \pm 3.66}$ | $86.42 \pm 5.05$ | $85.03 \pm 6.10$ | $85.83 \pm 1.61$ | - |
| I,B | $88.99 \pm 1.32$ | $83.3 \pm 2.74$ | $\mathbf{90.64 \pm 2.86}$ | $87.86 \pm 2.98$ | $86.76 \pm 1.31$ | $82.0 \times 10^{-6}$ |
| I,C | $\mathbf{89.72 \pm 1.26}$ | $85.09 \pm 2.21$ | $90.37 \pm 2.14$ | $\mathbf{89.29 \pm 1.93}$ | $\mathbf{87.61 \pm 1.23}$ | $\mathbf{9.54 \times 10^{-6}}$ |
|  | $\mathbf{AP_{50}}$ (%) | $\mathbf{AP_{60}}$ (%) | $\mathbf{AR_{50}}$ (%) | $\mathbf{AR_{60}}$ (%) | $\mathbf{AR_{70}}$ (%) | p-value |
| II,A | $80.26 \pm 5.62$ | $73.73 \pm 6.03$ | $89.23 \pm 3.85$ | $83.62 \pm 4.12$ | $64.96 \pm 4.39$ | - |
| II,B | $80.04 \pm 4.60$ | $72.37 \pm 5.14$ | $90.18 \pm 3.52$ | $83.52 \pm 4.10$ | $66.24 \pm 3.50$ | $0.985$ |
| II,C | $\mathbf{88.75 \pm 3.21}$ | $\mathbf{84.80 \pm 3.72}$ | $\mathbf{95.02 \pm 2.15}$ | $\mathbf{91.55 \pm 2.49}$ | $\mathbf{78.08 \pm 2.96}$ | $\mathbf{4.825 \times 10^{-4}}$ |

Table 1: Relevant metrics on 4 experiments: I: Nodule detection task. II: Nodule localization task. A: Without synthetic images in train set. B: With SPADE images in train set. C: With diffusion images in train set. p-value is generated between A (control experiment) and other experiments (B or C). AP and AR are Average Precision and Recall, and the subscript denotes the IoU% used.

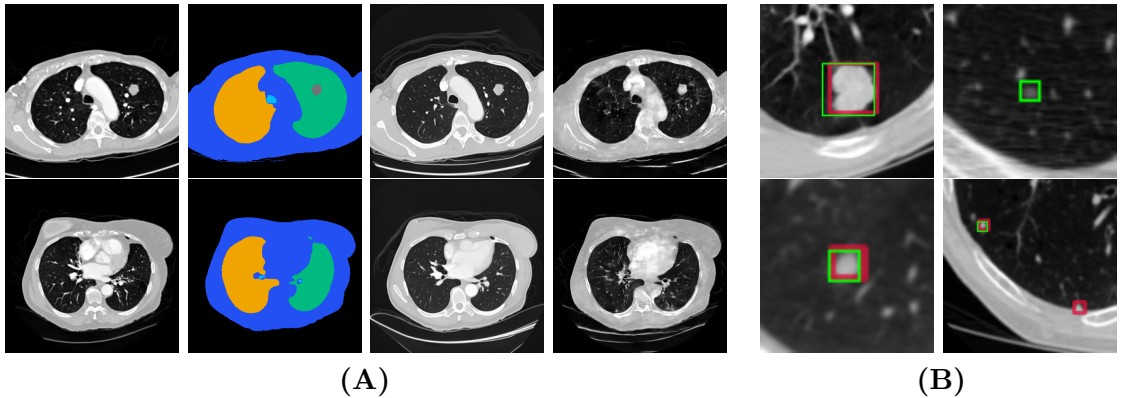

**(A)**          **(B)**

Figure 1: (A) Example images generated by SDM and SPADE. L-to-R: CT image, CT mask, SDM image, SPADE image. Top: Nodule slice. Bottom: Nodule-free slice. (B) Localization downstream task. Top: SDM, Bottom: SPADE. Left: Correctly identified nodules, Right: False Negative/Positive detections. Green box is ground truth and red box is prediction.

## 4. Discussion and Conclusion

The FID score of SDM-generated images is much lower than SPADE-generated images, indicating the quality of synthetic images via SDM is significantly better than synthetic images via SPADE. However, this comes at a trade-off where generating synthetic images using SDM is much more time-consuming and computationally expensive compared to SPADE (i.e. ~10 min/image for SDM whilst 320 images/min for SPADE running on Nvidia A6000 GPU). Surprisingly, in the SDM images, fine details in the trachea region of original images are preserved, while in the SPADE images the area is filled with random noises/strokes. Overall, our experiments have shown that SDM has the potential of generating high-fidelity pulmonary CT images, even with nodules of small diameters, as evident by the improvement of downstream tasks compared to SPADE and baseline. Future work include training SDM in 2.5D in order to perform 3D volume generation, and also to extend the mask class to include nodule malignancy.

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
