# OpenReview forum: "High-Fidelity Image Synthesis from Pulmonary Nodule Lesion Maps using Semantic Diffusion Model"
_MIDL.io/2023/Short_Paper_Track — MIDL 2023 Short paper track Poster_

### Official Review · Reviewer_K3bt · 2023-04-22
**Diffusion model for image synthesis**

**Rating:** 5
**Confidence:** 5

**Review:**

This paper explored the use of Semantic Diffusion Models (SDM) to generate high-fidelity pulmonary CT images from segmentation maps. Two common clinical downstream tasks including nodule detection and nodule localization are evaluated. Although it's interesting to see the adoption of diffusion models, the methodological details are missing and the experimental part is short of extensive comparison with existing methods such as GAN-based image synthesis methods, which weaken the contributions from this paper. I would suggest authors can perform in-depth method analysis and comparison.

---

### Official Review · Reviewer_nLLM · 2023-04-24
**Effective fancy data augmentation scheme**

**Rating:** 8
**Confidence:** 4

**Review:**

This paper suggest to use stable diffusion, and other image generation methods, as a data-augmentation scheme for pulmonary nodule lesion segmentation. This is a difficult task, with very small nodules, and good quality annotations is hard to come-by.

While the manuscript could be improved (I think), it remains relatively straightforward to follow, and the authors evaluate both performances in segmentation, and downstream task (localization).

The results show quite clearly that the task at hand benefit from the augmentation proposed. What remains unclear to me, is if the A setting has some data augmentation (non-learned, traditionnal) or not. But that is not an issue for a short paper, merely a suggestion to clarify this point for the poster.